# THE PROGRAM TESTING ABILITY OF LARGE LANGUAGE MODELS FOR CODE

## ABSTRACT

Recent development of large language models (LLMs) for code like CodeX and CodeT5+ demonstrates tremendous promise in achieving code intelligence. Their ability of synthesizing code that completes a program for performing a pre-defined task has been intensively tested and verified on benchmark datasets including HumanEval and MBPP. Yet, evaluation of these LLMs from more perspectives (than just program synthesis) is also anticipated, considering their broad scope of applications in software engineering. In this paper, we explore the ability of LLMs for testing programs/code. By performing thorough analyses of recent LLMs for code in program testing, we show a series of intriguing properties of these models and demonstrate how program testing ability of LLMs can be improved. Following recent work which utilizes generated test cases to enhance program synthesis, we further leverage our findings in improving the quality of the synthesized programs and show +11.77% and +4.22% higher code pass rates on HumanEval+ comparing with the GPT-3.5-turbo baseline and the recent state-of-the-art, respectively.

## 1 INTRODUCTION

The community has witnessed a surge in the development of large language models (LLMs), which have achieved incredible ability in understanding and generating not only texts but also code. LLMs for code (CodeX (Chen et al., 2021), StarCoder (Li et al., 2023b), CodeT5+ (Wang et al., 2023b), etc) have been widely adopted to a variety of applications to achieve code intelligence. However, current evaluation of these LLMs mostly focuses on program completion/synthesis, despite the models can also be utilized in other applications. As the field continues to advance, evaluation of these models from more perspectives is anticipated, which could facilitate deeper understanding of the LLMs.

The ability of automatically generating proper test suites is of great desire to software engineering, yet challenging. Being learning-based or not, current test generation efforts (e.g., fuzzing) primarily focus on creating diverse test inputs to identify faults in the code as much as possible via maximizing their coverage, e.g., line coverage and branch coverage (Fioraldi et al., 2020; Tufano et al., 2022; Dinella et al., 2022; Lemieux et al., 2023; Xia et al., 2023). Although such test inputs try to verify the (non-)existence of crashes and hangs of the tested code, they lack the ability of determining whether the code adheres to the aim of the function which is represented by input-output relationships. Automatic test case generation for this aim not only requires an high coverage of the code being tested but also necessitates a correct understanding of the "true" desired input-output relationships in the tested code, leaving it a challenging open problem.

Being capable of synthesizing correct code implementations given docstrings, LLMs for code seem capable of understanding the desired input-output relationship of a function described in natural language. This capability inspires applying these LLMs to generating test cases automatically (Chen et al., 2021). However, the ability of these models for program testing has not been systematically evaluated. In this paper, we systematically compare the ability of recent LLMs for code in testing from two perspectives focusing on both the correctness and diversity of the test cases, considering that 1) program testing is of great interest in software engineering and software security as mentioned and 2) automatically generated test cases can further be adopted to improve the program synthesis performance (Chen et al., 2023). Our analyses focus on algorithmic coding, based on the popular 164 problems from HumanEval+ (Liu et al., 2023a) and 427 sanitized problems from MBPP (Austin et al., 2021). It is worth noting that the model may encounter various scenarios when generating test

cases. It may generate test cases when provided with only natural language descriptions of the desire of the program, or it could generate test cases when given an "optimal" oracle implementation. In more complex situations, it may even need to test its own imperfect generated code or the code generated by other models. We consider 4 test-case generation settings (i.e., "self-generated" which uses each LLM to test code synthesized by the LLM itself, "all-generated" which lets all LLMs to test the same code synthesized by a group of four LLMs , "oracle" which tests an oracle implementation, and the "placeholder" in Figure 1) and test a collection of 11 competitive LLMs for code. We conducted a variety of experiments, from which intriguing takeaway messages are delivered.

As previously mentioned, several very recent papers (Shi et al., 2022; Li et al., 2023a; Chen et al., 2023) have shown that appropriate usage of generated test cases can improve the quality of program synthesis. Yet, the quality of generated test cases largely impacts the performance of such methods. Due to the lack of systematic evaluation of the testing ability of LLMs for code, it is unclear how to craft test cases that could be potentially more helpful to program synthesis. The studies in this paper also shed light on this. We will show that, substantially improved program synthesis performance can be gained by utilizing takeaway messages in our studies. Specifically, we can achieve +11.77% higher code pass rate on HumanEval+, in comparison with the GPT-3.5-turbo baseline. Compared with a very recent state-of-the-art called CodeT, our solution gains +4.22% higher code pass rate.

## 2 EVALUATION METRICS

To make the evaluation more reliable and comprehensive, it is crucial to first design some suitable metrics, like BLEU (Papineni et al., 2002), ROUGE (Lin, 2004), and the pass rate (Chen et al., 2021) for evaluating machine translation, text summarization, and program synthesis, respectively. In this section, we specify two main evaluation metrics to evaluate the program testing ability of LLMs, from the perspective of correctness and diversity.

**Pass rate** In software engineering, we expect test cases to represent some desired "ground-truth" functionality of the tested program/code. In practice, such "ground-truth" functionality can be described in the header comments of a function (i.e., docstrings of the function) and tested using the oracle implementation, as in HumanEval (Chen et al., 2021) and MBPP Austin et al. (2021). The oracle program/code should be able to pass the test, if a generated test case is correct. Therefore, we leverage the pass rate as a measure to evaluate the correctness of the generated test cases. For a fair comparison, we instruct each model to generate three test cases in the prompt, and, when a model generates more than three test cases, we select the first three for evaluation. Assuming that there are in total $M$ programming problems in an experimental dataset and, for each problem, we have $N$ program/code implementations to be generated test cases for. Each model has only one chance to generate these test cases for each program/code. Then, we calculate the pass rate as:

$$P = \frac{1}{MN} \sum_{i=1}^{M} \sum_{j=1}^{N} \frac{p_{ij}}{n_{ij}}, \tag{1}$$

where $n_{ij}$ is the number of test cases in $\mathcal{Q}_{ij}$ which includes no more than three test cases generated for the $j$-th program/code implementation of the $i$-th problem by the evaluated LLM at once, i.e., $\mathcal{Q}_{ij} = \{(x_{ijk}, y_{ijk})\}_k$, and $p_{ij}$ is the number of test cases (in $\mathcal{Q}_{ij}$) that do not fail the oracle.

The pass rate defined in Eq. (1) measures correctness of the generated test cases. However, as can be seen in Figure 1, the model can generate duplicate test cases that are less helpful, even though they are correct. To avoid such an evaluation bias, we further advocate deduplication in the set of test cases that are considered as correct, which leads to computation of a deduplicated pass rate defined as $P' = \frac{1}{MN} \sum \sum p'_{ij}/n'_{ij}$, where we use $'$ to denote the numbers of unique test cases.

**Coverage rate** In addition to the above pass rates, we further consider coverage rate as a more fine-grained metric for evaluating the diversity of the generated test cases. According to its definition, converge rate computes the degree to which the code is executed, given a test case. Since, for each program/code, we keep no more than three test cases at once, we calculate how much percentage of the control structure is covered given these test cases. Similar to Eq. (1), we evaluate the performance of testing all programs/code over all $M \times N$ times of generation, i.e., we calculate

$$C = \frac{1}{MN} \sum_{i=1}^{M} \sum_{j=1}^{N} c_{ij}, \tag{2}$$

where $c_{ij}$ is the per-test-case branch coverage rate. We apply the *pytest* [1] library to evaluate the branch coverage for all the three test cases for each code and average the results for all programs/code and all problems. Apparently, $C \leq 1$, and a higher $C$ shows better testing ability of an LLM, since we expect all parts of the programs/code to be executed to find our all potential bugs.

# 3 LARGE LANGUAGE MODELS FOR CODE

In this section, we outline the evaluated models. We adopt some "small" models whose numbers of parameters are around 1B (to be more specific, from 770M to 1.3B in our choices) and some larger models that achieve state-of-the-art performance in the task of program synthesis.

For the small models, we use **InCoder** (1.3B) (Fried et al., 2023), **CodeGen2** (1B) (Nijkamp et al., 2023a), **CodeT5+** (770M) (Wang et al., 2023b), and **SantaCoder** (1.1B) (Allal et al., 2023). In-Coder is a unified generative model that can perform program/code synthesis as well as code editing, and it combines the strengths of causal language modeling and masked language modeling. The CodeGen2 model was trained on a deduplicated subset of the Stack v1.1 dataset (Kocetkov et al., 2023), and its training is formatted with a mixture of objectives for causal language modeling and span corruption. CodeT5+ is an encoder-decoder model trained on several pre-training tasks including span denoising and two variants of causal language modeling. SantaCoder was trained on the Python, Java, and JavaScript code in the Stack dataset. The pass rate (Chen et al., 2021) of programs generated by these models is compared in Table 1. When evaluating the (program) pass rate, we let the model generate 200 code implementations for each problem, and we set the temperature to 0.2, 0.6, and 0.8 for calculating pass@1, pass@10, and pass@100, respectively.

As for larger models that achieve state-of-the-art program synthesis performance, we use **CodeGen2** (16B) (Nijkamp et al., 2023a), **CodeGen-Multi** (16B) Nijkamp et al. (2023b), **CodeGen-Mono** (16B) Nijkamp et al. (2023b), **StarCoder** (15B) (Li et al., 2023b), **WizardCoder** (15B) (Luo et al., 2023), **CodeGeeX2** (6B) (Zheng et al., 2023), and **GPT-3.5-turbo**. CodeGen-Multi and CodeGen-Mono are two large models from the first version of CodeGen. CodeGen-Multi was first trained on the pile dataset (Gao et al., 2020) and then trained on a subset of the publicly available BigQuery dataset which contains code written in C, C++, Go, Java, JavaScript, and Python. Based on the 16B CodeGen-Multi model, CodeGen-Mono (16B) was obtained by further tuning on a set of Python code collected from GitHub. Given a base model that was pre-trained on 1 trillion tokens from the Stack dataset, the 15B StarCoder model was obtained by training it on 35B tokens of Python code. WizardCoder further empowers StarCoder with instruction tuning, following a similar instruction evolution strategy as in WizardLM (Xu et al., 2023). CodeGeeX2, the second generation of a multi-lingual generative model for code, is implemented based on the ChatGLM2 architecture and trained on more code data. GPT-3.5-turbo is a very capable commercial LLM developed by OpenAI and we accessed it in August, 2023. For these large LLMs, we tested pass@1 of all models except GPT-3.5-turbo (whose result can be directly collected from Liu et al. (2023a)'s paper). By sorting their pass@1 from high to low, they are ranked as: GPT-3.5-turbo (61.7%), WizardCoder (46.23%, 15B), CodeGeeX2 (29.97%, 6B), StarCoder (27.9%, 15B), CodeGen-Mono (26.15%, 16B), CodeGen2 (19.33%, 16B), CodeGen-Multi (15.35%, 16B). The ranks on the MBPP dataset are similar.

# 4 CODE TO BE TESTED

For evaluating the testing ability of LLMs, we need an oracle to express the ground-truth functionality of the tested code. Fortunately, current datasets for evaluating program synthesis performance often provide such oracles (see HuamnEval (Chen et al., 2021) and MBPP (Austin et al., 2021)). In our experiments, we utilize an amended version of HumanEval called HumanEval+ (Liu et al., 2023a), together with MBPP (the sanitized version). These datasets are established to evaluate basic Python programming performance of LLMs, and they contain 164 and 427 problems, respectively.

## 4.1 IMPERFECT CODE IMPLEMENTATIONS

In order to simulate real-world scenarios where the tested code is often buggy, we first adopt synthesized programs/code as the programs/code to be tested, considering that the synthesis of even

---

[1] https://pytest.org

Figure 1: Testing for (a) self-generated code, (b) all-generated code, (c) oracle, and (d) placeholder.

| Model | Size | Pass@1 | Pass@10 | Pass@100 |
|-------|------|--------|---------|----------|
| InCoder | 1.3B | 6.99%/14.06% | 14.20%/34.98% | 23.76%/55.34% |
| CodeGen2 | 1B | 9.19%/17.50% | 16.06%/36.86% | 25.90%/59.32% |
| CodeT5+ | 770M | 12.95%/28.02% | 25.09%/47.69% | 37.56%/65.26% |
| SantaCoder | 1.1B | 15.21%/29.42% | 26.01%/51.30% | 43.80%/69.10% |

Table 1: *Program synthesis performance* of the *small* LLMs (whose number of parameters is around 1 billion) evaluated on HumanEval+ / MBPP (sanitized).

state-of-the-art LLMs is still imperfect. We evaluate the performance of each LLM in testing code that was generated by itself (which is denoted as "**Self-generated**") and code in a set consisting of program completion results of several different LLMs (which is denoted by "**All-generated**"). That said, the compared LLMs take different code implementations when generating test cases for each programming problem in the self-generated setting. Whereas, in the all-generated setting, the same program/code implementations are given to different LLMs for generating test cases for comparison. In practice, we apply InCoder (1.3B), CodeGen2 (1B), CodeT5+ (770M), and SantaCoder (1.1B) to construct the all-generated program/code set, while, in the self-generated setting, each LLM first synthesize code and complete a program to fulfill the requirement of each programming problem, and the LLM then generates test cases with the synthesized programs/code in its prompts. The temperature for all LLMs is uniformly set to 0.2 for synthesizing the programs/code in both settings. We obtain 100 program/code completions for each problem and we prompt each LLM to generate 3 test cases for every program/code implementation in the self-generated setting, and we sampled 100 implementations from the synthesis results of InCoder (1.3B), CodeGen2 (1B), CodeT5+ (770M), and SantaCoder (1.1B) to form the all-generated code set, i.e., we have $N = 100$ for these settings.

We follow the same way of generating code as introduced in the papers of these LLMs. For model without instruction tuning, like InCoder and CodeT5+, we synthesize programs/code using the default prompt given by each programming problem in the test dataset, while, for models that have adopted instruction tuning, e.g., WizardCoder, we use the recommended prompt in their papers.

## 4.2 OPTIMAL CODE IMPLEMENTATIONS (ORACLE)

As a reference, we also report the performance of generating accurate and diverse test cases when the written code is perfectly correct, which is achieved by adopting the oracle as the programs/code to be tested (and such a setting is denoted by "**Oracle**"). Since Liu et al. (2023a) have reported that some oracle code in the HumanEval dataset can be incorrect, we adopt the amended oracle set in HumanEval+ in this setting. We further used the revised oracle code implementations instead of the original ones in evaluating the pass rate (i.e., $P'$) of the generated test cases. Considering that the public datasets often only provide one oracle implementation for each problem, and to keep the uncertainty of evaluation results consistent, we copy the oracle implementation by $100\times$ and we

prompt to generate 3 test cases for each of these copies. It can be regarded as letting $N = 100$, just like in the previous settings in Section 4.1.

### 4.3 No Implementation (**Placeholder**)

In certain scenarios, we require test cases before the function/program has been fully implemented, hence we also evaluate in a setting where the main body of a tested function/program is merely a placeholder, as depicted in Figure 1(b). This scenario often occurs when the main code has not yet been implemented for a function/program or the test engineer does not want to introduce implementation bias to the LLM when generating test cases for a function/program. We denote such a setting as "**Placeholder**" in this paper. We also let $N = 100$, as in the oracle setting.

## 5 Test Case Generation

In this section, we introduce how test cases can be generated, when the implementation of a function/program is given as described in Section 4. In this paper, a desired test case is a pair of input and its expected output for the function/program defined in the context. As an example, Figure 1 demonstrates some test cases for the programming problem of checking whether the two words satisfy a specific rotation pattern. To generate test cases, we use the LLMs introduced in Section 3.

We wrote extra prompts to instruct the LLMs to generate three test cases for each given code which include docstrings that describe the purpose of this function, as depicted in Figure 1. Our instruction commands the LLMs (1) to "check the correctness of this function with three test" and (2) to start writing test code with an "`assert`" statement and the tested function, which specifies the format of the test cases as input-output pairs that can be parsed. For instance, given the example in Figure 1, the extra prompt should be "`# Check the correctness of this function with three test cases \n assert cycpattern_check`".

We then concatenate the extra prompt with the code and feed the concatenation into each LLM, for extracting test cases from the model output. The LLM will try to complete the given input by generating one or more "`assert`" statement(s), and we split the generation results into sub-strings, with "`assert`" as the separator. Each sub-string is then considered as a test statement, and we only take the first three statements if there exist more than three statements, as has been introduced in Section 2. Such a split can be considered as an effective post-processing operation which largely improves the quality of the generated test code, considering that some non-sense code pieces may be generated in the output of the LLMs. When using HumanEval+ and MBPP, we try removing test cases in the docstrings of the function, if there exist any, just to get rid of the broad hints from the docstrings (Chen et al., 2023). The temperature for generating test cases is kept as 0.2.

Once obtained, the generated test cases are then compiled, and evaluated for their correctness and diversity to report the pass rate $P'$ and the coverage rate $C$. When calculating, for each problem and every set of completions generated, we create a temporary folder.

## 6 Main Results for Test Case Generation

The experiment results of small and large LLMs on HumanEval+ can be found in Table 2 and Table 3, respectively. Table 4 shows the results on MBPP. There are several takeaways from these tables.

- **First**, the test cases generated by LLMs can show a descent pass rate, and this pass rate is even higher than the code pass rate on HumanEval+, which holds for both large and small

| Model | Size | Oracle | Self-generated | All-generated | Placeholder |
|-------|------|--------|----------------|---------------|-------------|
| InCoder | 1.3B | 21.31% (61.43%) | 23.37% (59.36%) | 22.72% (61.10%) | 25.19% (62.75%) |
| CodeGen2 | 1B | 31.63% (**71.55%**) | 30.62% (69.38%) | 30.93% (69.70%) | 30.69% (69.00%) |
| CodeT5+ | 770M | **35.43%** (71.45%) | **32.34%** (70.45%) | **31.49%** (69.75%) | **32.67%** (70.67%) |
| SantaCoder | 1.1B | 30.97% (71.46%) | 30.43% (**70.81%**) | 30.13% (**70.55%**) | 30.78% (**71.24%**s) |

Table 2: The pass rates (and coverage rate) of the test cases generated on HumanEval+ in different settings for LLMs with around 1 billion parameters.

| Model | Size | Oracle | Self-generated | All-generated | Placeholder |
|---|---|---|---|---|---|
| CodeGen-Multi | 16B | 43.88% (67.91%) | 41.85% (69.30%) | 40.38% (66.97%) | 39.74% (68.28%) |
| CodeGen2 | 16B | 46.34% (73.07%) | 45.44% (73.17%) | 42.00% (72.45%) | 42.69% (72.86%) |
| CodeGen-Mono | 16B | 49.03% (74.82%) | 45.73% (73.74%) | 43.91% (73.66%) | 44.92% (73.63%) |
| StarCoder | 15B | 55.07% (76.02%) | 52.52% (72.45%) | 48.20% (72.30%) | 50.58% (74.52%) |
| CodeGeeX2 | 6B | 57.03% (74.42%) | 53.16% (73.55%) | 49.28% (70.32%) | 51.78% (73.08%) |
| WizardCoder | 15B | 53.89% (**77.87%**) | 55.47% (76.07%) | 48.02% (**75.27%**) | 49.89% (**75.12%**) |
| GPT-3.5-turbo | - | **71.03%** (77.85%) | **72.45%** (77.24%) | **59.24%** (74.99%) | **66.28%** (74.03%) |

Table 3: The pass rates (and coverage rate) of the test cases generated on HumanEval+ in different settings for LLMs whose parameters are obviously more than 1 billion.

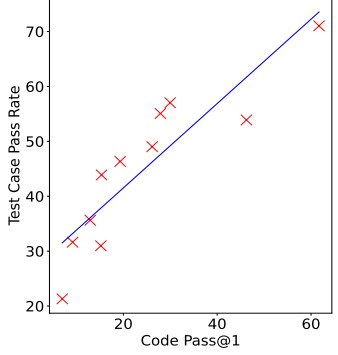

Figure 2: The correlation between code past rate and test pass rate in the "Oracle" setting.

Figure 3: How the correctness of the test cases changes with their order when being generated.

LLMs. Such a result is consistent with intuitions from previous work which rejects code that cannot pass the generated tests to improve the quality of program synthesis.

- **Second**, the correctness of the generated test cases is positively correlated with the LLM's ability of generating code (see Figure 2, where each red cross represents the performance of a model), which means an LLM showing the state-of-the-art program synthesis performance is possibly also the state-of-the-art LLM for program testing. As shown in Tables 2 and 3, GPT-3.5-turbo, which synthesizes programs/code with the highest correctness, provides test cases with the highest pass rate (71.03%) on HumanEval+. For an LLM, the more accurate it is capable of synthesizing programs/code on a dataset, the more powerful testing ability will probably be exhibited on the same dataset. There also exist a few exceptions, e.g., SantaCoder (1.1B) outperforms CodeT5+ (770M) and CodeGen2 (1B) in generating code, but it shows inferior performance in program testing on HumanEval+. By carefully examining the test cases yielded by SantaCoder on HumanEval+, we found that it tends to generate more complex and longer test cases than CodeT5+ for several problems on HumanEval+, which are often more desirable in program testing. This is also why the SantaCoder test cases show higher coverage rates in Table 2. To be concrete, in Problem 131 in HumanEval+, where the program is required to return the product of all digits with an odd position in a positive integer $n$ (which is the input), the test input provided by CodeT5+ tends to be small for this problem, e.g., $n = 2$, while the SantaCoder test cases tend to have more digits (e.g., $n = 12358$), which is helpful in digging out hidden bugs. Yet, generating longer and more complex test cases is more challenging, and the correctness can be lower.

- **Third**, as can be seen in Tables 3 and 4, generating test cases using *large* LLMs with their self-generated code (in the prompts) often leads to a higher level of correctness, compared with the placeholder results. This observation is in fact unsurprising, considering that generating code first and test case afterwards resembles the chain-of-thought prompting (Wei et al., 2022) (if adopting the placeholder is regarded as a plain prompting), which is beneficial to reasoning. Moreover, the self-generated performance of an LLM sometimes even outperforms its testing performance with an oracle, and we ascribe this to: 1) randomness in the style of the oracles which are few in number and/or 2) less distribution shift between self-generated code in prompt and the training code, for some powerful LLMs.

| Model | Size | Oracle | Self-generated | All-generated | Placeholder |
|---|---|---|---|---|---|
| InCoder | 1.3B | 21.56% (46.81%) | 17.98% (46.11%) | 19.53% (46.45%) | 22.58% (46.72%) |
| CodeGen2 | 1B | 25.61% (54.26%) | 21.85% (53.09%) | 23.15% (50.43%) | 22.81% (52.11%) |
| CodeT5+ | 770M | 29.02% (**56.86%**) | 24.44% (52.31%) | 24.84% (**53.20%**) | 25.59% (**55.81%**) |
| SantaCoder | 1.1B | **32.37%** (55.68%) | **26.40%** (**52.38%**) | **26.20%** (52.83%) | **26.53%** (53.86%) |
| CodeGen-Multi | 16B | 41.32% (60.63%) | 35.96% (59.03%) | 34.17%,(58.09%) | 34.84% (58.92%) |
| CodeGen2 | 16B | 45.30% (62.15%) | 38.67% (60.16%) | 36.77% (58.59%) | 37.27% (59.16%) |
| CodeGen-Mono | 16B | 50.24% (64.39%) | 43.94% (62.94%) | 39.55% (61.99%) | 42.41% (62.31%) |
| StarCoder | 15B | 54.84% (65.10%) | 46.77% (63.60%) | 42.80% (61.95%) | 45.35% (62.66%) |
| CodeGeeX2 | 6B | 52.45% (64.64%) | 44.52% (63.72%) | 41.72% (60.48%) | 43.86%,(63.51%) |
| WizardCoder | 15B | 57.85% (**66.68%**) | 46.56% (64.86%) | 41.62% (60.72%) | 47.45% (64.54%) |
| GPT-3.5-turbo | - | **74.30%** (66.19%) | **66.14%** (**65.30%**) | **49.56%** (**62.95%**) | **63.34%** (**64.72%**) |

Table 4: The pass rates (and coverage rate) of the test cases generated on MBPP.

- **Fourth**, with only a few exception, test cases obtained using the oracle code exhibit slightly higher code coverage, while the coverage rate achieved in the other settings (i.e., the self-generated, all-generated, and the placeholder settings) is often slightly lower.

The above four takeaway messages can all be inferred from Tables 2, 3, and 4. In addition to all these results, we conduct more experiments to achieve the following takeaway messages.

- **Fifth**, by analyzing the relationship between the quality of code in prompts and the correctness of test, we found that correct code implementation in the prompt often leads to higher quality of test code generation than the case when some incorrect code is given. We conducted an experiments where we first select programming problems in HumanEval+, where the code pass rate of an LLM is neither 0% or 100%. Then we separate self-generated programs/code of the model into two groups, with one group only contains programs/code that are considered as correct and the other only contains incorrect programs/code. In Table 5, we compare the performance of using these two sorts of code in the prompt, for generating test cases using the same LLM. Apparently, the quality of test cases obtained with correct programs/code is obviously higher. We further evaluate the overall testing performance of LLMs with only correct self-generated programs/code, if there exists any, in their prompts. Unlike in Table 5 where we do not take problems that can be 100% or 0% solved, we take all given problems in this evaluation, except, for every problem, we eliminate all incorrect self-generated programs/code if there exist at least one correct implementation synthesized by the evaluated LLM. By doing so, we can observe substantially improved program testing ability on HumanEval+ (i.e., 74.95% for GPT-3.5-turbo, 56.87% for WizardCoder, 54.33% for CodeGeeX2, and 53.24% for StarCoder), comparing with the original self-generated results in Table 3. The same on MBPP.

- **Sixth**, by conducting an additional experiment, we further compare the quality of test cases collected from different positions in the generation results. For every set of the three generated test cases, we analyze the relationship between their correctness and the order when they are generated. The results are illustrated in Figure 3. As can be seen in the figure, the first generated test case often shows the best correctness and the latterly generated ones are more incorrect. This may be due to the fact that the model tends to first generate content with a high level of confidence (which is also more likely to be correct).

## 7 IMPROVING PROGRAM SYNTHESIS USING THE GENERATED TEST CASES

High quality test cases are not only desired in program analyses, but also helpful to program synthesis. Previous methods have successfully used generated test cases to improve the performance of LLMs in synthesizing programs/code. For instance, Li et al. (2023a) designed a special prompt which involves the test cases as an preliminary, if they are available, for generating programs/code. One step further, Chen et al. (2023) proposed CodeT, which leverages the LLM to obtain test cases first and tests all synthesized programs/code with these test cases by performing a dual execution agreement, and it picks the code in the largest consensus set (i.e., the consensus set with the most code implementations and test cases) as output to obtain state-of-the-arts program synsthesis performance. We encourage interested reader to read the original paper.

| Model | Size | w/ correct code | w/ incorrect code | #Problem |
|---|---|---|---|---|
| InCoder | 1.3B | **28.55%** | 27.39% | 27 |
| CodeGen2 | 1B | **27.25%** | 25.74% | 11 |
| CodeT5+ | 770M | **40.19%** | 36.78% | 27 |
| SantaCoder | 1.1B | **37.45%** | 34.08% | 24 |
| CodeGen-Multi | 16B | **55.49%** | 50.06% | 32 |
| CodeGen2 | 16B | **43.56%** | 39.31% | 29 |
| CodeGen-Mono | 16B | **45.18%** | 42.86% | 56 |
| StarCoder | 15B | **58.16%** | 57.08% | 68 |
| CodeGeeX2 | 6B | **52.84%** | 48.63% | 51 |
| WizardCoder | 15B | **48.02%** | 45.12% | 54 |
| GPT-3.5-turbo | - | **75.39%** | 68.52% | 126 |

Table 5: With the correct (self-generated) code, the LLMs show stronger ability of generating correct test cases on HumanEval+ (evluated only on those problems that can neither be 0% solved nor 100% solved), than in the case where incorrect self-generated code is given in the prompts. Since most LLMs cannot generate any correct code for many hard problems while they often generate incorrect code even for easy problems, the number of tested problems in this experiment increases with the power of the tested LLM, as shown in the rightmost column.

In the previous section, we have obtained results about many intriguing properties of the program testing performance of LLMs for code. In this section, we would like to drive the readers to think whether it is possible to utilize these results to improve the program synthesis performance, considering that the test cases (hand-crafted and given or automatically generated in particular) are widely and successfully used in program synthesis. We shall demonstrate that, by utilizing takeaway messages in Section 6, the program synthesis performance of previous methods can be improved significantly. Taking CodeT as an example of the previous state-of-the-art, the method uses a placeholder to generate test cases and treats all the test cases as equally correct as a prior. However, as discussed in our third takeaway message, using self-generated code helps to achieve more powerful ability in generating correct test cases. Moreover, if multiple test cases are provided in a single run of generation given an LLM, the correctness of the test cases decreases with their generation order, as shown in our fifth point. Hence, to obtain superior program synthesis performance, we introduce two simple modifications to it: 1) we employ the "self-generated" setting instead of the "placeholder" setting for generating test cases, which means we utilized synthesize programs in prompts when generating test cases for each program, 2) we assign different weights to the generated test cases based on their order in each generation result, which means we used the rank of each generated test case to re-weight its contribution to the consensus set it belongs to.

We test the effectiveness of using 1) the prompt which involves self-generated (SG) code as the test cases generated in this setting show higher correctness than the baseline placeholder setting and 2) the rank-based re-weighted (RW) test cases, in improving program synthesis performance on HumanEval+. Following Chen et al. (2023), we used a temperature of 0.8 to generate code and self-generated test cases. After obtaining the consensus set, we re-weight test case by $p^{i-1}$ with $i$ being its order in the model output, and we let $p = 0.8$. That is, instead of directly using their counting numbers, we use the sum of $p^{i-1}$ and the final score of a consensus set is then the sum of a) $\sum p^{i-1}$ and b) the number of code implementations in the consensus set, and code implementations in the consensus set with the highest score are considered as the best solutions.

Table 6 shows the results. We compare CodeT with CodeT+SG, CodeT+RW, and CodeT+SG+RW. For CodeT, we follow their official implementation and generate $100 \times 5$ test cases for each problem. For fair comparison, we ensure that our solutions with SR and/or RW generate the same numbers of program implementations and test cases as CodeT does. Hence, for each problem in HumanEval+, we synthesize a program together with its 5 test cases for 100 times when SR and/or RW are incorporated, i.e., we have $i \in \{1, 2, 3, 4, 5\}$. It can be seen from the table that both SG and WR improves the program synthesis performance considerably on most LLMs, except for Incoder, CodeGen2-1B, CodeT5+, and SantaCoder for which the test cases generated in the placeholder setting show similar or even higher correctness than in the self-generated setting and SG fails with them. For some LLMs, SG is more powerful, while, on the other models including SantaCoder and StarCoder, RW is more powerful. By combining SG and RW, the program synthesis performance of most powerful LLMs in Table 6 improves, comparing to only using one of the two. On GPT-3.5-turbo and WizardCoder, which are the best two models in synthesizing programs on HumanEval+, we achieve +4.22% and +3.04% performance gains for CodeT, respectively, with SG & RW.

| Model | Size | Baseline | CodeT | + SG | + RW | + SG & RW |
|-------|------|----------|-------|------|------|-----------|
| InCoder | 1.3B | 6.99% | 9.85% | 9.45% | **10.26%** | 9.98% |
| CodeGen2 | 1B | 9.19% | 15.15% | 14.89% | **15.67%** | 15.35% |
| CodeT5+ | 770M | 12.95% | 16.57% | 16.28% | **17.19%** | 16.98% |
| SantaCoder | 1.1B | 15.21% | 18.43% | 18.17% | **18.75%** | 18.63% |
| CodeGen-Multi | 16B | 15.35% | 24.50% | 25.71% | 25.72% | **26.95%** |
| CodeGen2 | 16B | 19.33% | 27.56% | 28.51% | 28.43% | **29.63%** |
| CodeGen-Mono | 16B | 26.15% | 35.63% | 36.69% | 36.63% | **37.95%** |
| StarCoder | 15B | 27.90% | 40.46% | 41.21% | 42.12% | **43.15%** |
| CodeGeeX2 | 6B | 29.97% | 44.16% | 45.23% | 44.92% | **46.32%** |
| WizardCoder | 15B | 46.23% | 58.41% | 60.13% | 59.60% | **61.45%** |
| GPT-3.5-turbo | - | 61.70% | 69.25% | 72.45% | 70.75% | **73.47%** |

Table 6: *Program synthesis performance* (Pass@1) of LLMs can be significantly improved by using our takeaway messages in Section 6. The experiment is on HumanEval+.

## 8 RELATED WORK

**Test case generation via program analysis.** Generating reasonable test cases for analyzing programs is a long standing problem in the software engineering community. Various program analysis techniques, e.g., fuzzing, have been developed for achieving this goal. AFL++ (Fioraldi et al., 2020) is the most popular tool which incorporate many techniques in this category. A major weakness of these techniques is understandability of the generated test cases.

**Test case generation via deep learning.** The invention of transformer and self-supervised pre-training have brought a breakthrough to programming language processing and program testing (Fioraldi et al., 2020; Tufano et al., 2022; Dinella et al., 2022). After being trained in a self-supervised manner on a large and diverse code corpus, LLMs have demonstrated remarkable abilities in understanding and synthesizing programs. We have also witnessed the adaptation of pre-trained LLMs (e.g., ChatGPT) to fuzzing (Xia et al., 2023) very recently. Similarly, Lemieux et al. (2023) utilized Codex to provide example test cases for under-covered functions, which prevents the coverage improvements stall. Nevertheless, there still lack and require in-depth analyses and intensive comparisons of different LLMs in program testing, considering that powerful LLMs emerge continuously. For instance, the recent WizardCoder (Luo et al., 2023) exhibits an obvious program synthesis superiority over other contemporary open-source LLMs. In our study, we focus on the analyses and comparison of the LLMs in writing test code and generating test cases.

**Evaluation of Large Language Model.** Recently, large language models (LLMs) has incited substantial interest in both academia and industry. In order to evaluate the capabilities of large language models, a variety of effort have been devoted from the perspectives of natural/programming language processing accuracy, robustness, ethics, biases, and trustworthiness, etc. For instance, PromptBench (Zhu et al., 2023) demonstrates that current LLMs are sensitive to adversarial prompts, and careful prompt engineering is necessary for achieving descent performance with them. Another example, DecodingTrust (Wang et al., 2023a), offers a multifaceted exploration of trustworthiness of the GPT models, especially GPT-3.5 and GPT-4. The evaluation expands beyond the typical trustworthiness concerns to include several new critical aspects. Agentbench (Liu et al., 2023b) evaluates LLM as agents on challenging tasks in interactive environments. Their experimental results show that, while top commercial LLMs present a strong ability of acting as agents in complex environments, there is a significant disparity in performance between them and their open-source competitors.

## 9 CONCLUSION

In this paper, we have performed thorough analyses of recent LLMs (mostly LLMs for code) in testing programs/code. Through comprehensive experiments with 11 LLMs on programming benchmark datasets including HumanEval+ and MBPP (the sanitized version), we have uncovered a range of intriguing characteristics of these LLMs for program/code testing. We have illustrated how the program testing capabilities of these LLMs can be enhanced in comparing intensive empirical results in four different settings. Based on our findings, we are also capable of improving the performance of state-of-the-art LLMs in synthesizing programs/code with test cases of higher quality. As a preliminary research work, we believe our paper can provide new research insights and spark new ideas in program/code synthesis, test-case generation, and LLM understanding, and we look forward to future exploration in this direction in future work.

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

| Model | Size | Self-generated | All-generated |
|-------|------|----------------|---------------|
| InCoder | 1.3B | 54.38% | 46.97% |
| CodeGen2 | 1B | 56.79% | 48.78% |
| CodeT5+ | 770M | 60.03% | 54.16% |
| SantaCoder | 1.1B | 56.58% | 54.42% |
| CodeGen-Multi | 16B | 53.09% | 51.27% |
| CodeGen2 | 16B | 55.66% | 53.11% |
| CodeGen-Mono | 16B | 57.62% | 58.05% |
| StarCoder | 15B | 60.29% | 55.09% |
| WizardCoder | 15B | 71.57% | 56.42% |
| GPT-3.5-turbo | - | 72.42% | 62.91% |

Table 7: The coverage rate of the test cases generated on HumanEval.

# A APPENDIX

## A.1 FURTHER ANALYSIS OF EXPERIMENTAL RESULTS

In this part, we provide further analysis of the experimental results in Section 6.

With regard to the situation where the test case quality generated by SantaCoder is lower than that generated by CodeT5+ on the HumanEval+ dataset, we have explained that this is probably because SantaCoder tends to generate longer and more complex test cases. Here we further demonstrate that SantaCoder is capable to generate more accuracy output when given the same testing input as that of CodeT5+'s. To show this, we first extract the input part of the test cases (which includes testing inputs paired with their corresponding outputs) generated by CodeT5+ in the oracle setting. We then let SantaCoder to generate testing outputs given these inputs, and assessed the accuracy of such test cases. The results show that, given these testing inputs already, SantaCoder and CodeT5+ obtain an correctness of **41.67%** and **40.34%**, respectively, showing that SantaCoder is indeed stronger, if the same testing input is given and it does not have the chance to yeild more complex testing inputs.

## A.2 ANALYSIS OF CODE COVERAGE

In the previous sections, when evaluating the code coverage of test cases, we used standard code as the reference. To further assess the code coverage ability of test cases generated by the model, we separately measured the coverage of test cases for their corresponding generated code. This involves measuring the coverage of self-generated test cases for self-generated code and the coverage of all-generated test cases for all-generated code. The results are shown in Table 7.

## A.3 THE INFLUENCE OF DIFFERENT PROMPTS

As mentioned in Section 5 in the paper, the prompt for generating test cases are given by concatenating the function definitions and docstrings ("def cycpattern_check(a, b): \n \t ""...."), the code implementation ("c=a \n ....") or a placeholder ("pass"), and a comment given to prompt test case generation ("# Check the correctness of this function with three test cases..."). In our early experiments, we found that modifying the final comment given to prompt test case generation only has a relatively small impact on the test case pass rate. We have tried e.g., "# Verify if the function is accurate and generate three test cases..." and "# Generate three test data to verify the correctness of this function..." and only observed less than $0.50\%$ difference in correctness of the obtained test cases.

