# OpenReview forum: "The Program Testing Ability of Large Language Models for Code"
_ICLR.cc/2024/Conference — ICLR 2024 Conference Withdrawn Submission_

### Official Review · Reviewer_PLSn · 2023-10-28

**Soundness:** 3 good
**Presentation:** 2 fair
**Contribution:** 2 fair
**Rating:** 3
**Confidence:** 4

**Summary:**

This work explores the ability of language language models (LLMs) for testing programs. Two benchmark datasets and 11 LLMs were evaluated. Results on pass rate and coverage were reported and discussed.

**Strengths:**

This paper addresses an important problem that is the testing ability of LLMs.
The writing is overall good, though there are some sentences that I am not sure about the meaning.
The results, while being preliminary, might be still useful.
10+ LLMs have been evaluated.

**Weaknesses:**

The significance and novelty of this work is unclear. The pass rate and test generation of the LLM has been already studied in existing work (Chen et al., 2021). Important relevant work was also missed in the paper. It was said, on Page 1, that “In this paper, we, for the first time, analyze the ability of recent LLMs in testing programs/code.” This is at least over-claiming. Existing work like CodeMosa already applies LLMs for improving code coverage and it was not mentioned or compared in this paper.

CODAMOSA: Escaping Coverage Plateaus in Test Generation with Pre-trained Large Language Models (ICSE 2023)

This is an empirical paper, and its evaluation is neither systematic nor comprehensive.

Two datasets were used in the evaluation. The selection of datasets seems to be arbitrary. What are their characteristics? Are they sufficient? Why are they representative? What will happen to programming languages other than Python?
The metrics for evaluating the experimental results cannot be taken as granted. The pass rate indicates the ratio of tests that pass unit testing, but it cannot tell if a test is good or not and often those tests that do not pass are regarded as more important as the aim of testing is to find defects in the code.

Code coverage is definitely more interesting than pass rate. The code coverage in the results look mediocre. Also, the computation of code coverage in Eq. (2) seems to be wrong. Different tests can cover the same branches and they are aggregated with considering the redundancy in Eq. (2). As a result, the final coverage might be over-estimated.

If the focus of this paper is exploring the testing ability of LLMs, the use of LLMs to generate the program/code shifts the focus and unnecessarily complicates the evaluation.

I do not understand what the following statement means: “the test cases generated by LLMs can show a descent pass rate, and this pass rate is even higher than the code pass rate on HumanEval+, which holds for both large and small LLMs.” (Page 5)

**Questions:**

1. Existing work already applies LLMs for improving code coverage. What’s the advantage of the approach in this paper? Will it result in on par to even higher coverage? If the answer is no, does it mean that testing ability of LLM exercised in this paper is not the state of the art?

2. What are the principles for dataset selection in the paper?

3. How to evaluate the quality of tests generated by LLMs in this paper?

---

> ### Author Response · Authors · 2023-11-21
>
> Thanks for your insightful comments! Please see our responses below.
>
> **Q1:** The pass rate and test generation of the LLM has been already studied in existing work (Chen et al., 2021). Important relevant work was also missed in the paper. It was said, on Page 1, that “In this paper, we, for the first time, analyze the ability of recent LLMs in testing programs/code.” This is at least over-claiming. Existing work like CodeMosa already applies LLMs for improving code coverage and it was not mentioned or compared in this paper.
>
> **A:** We appreciate the pointer to related work. In fact, comparing to previous work such as CodeMosa, we focus on a comprehensive study (including the correctness of test cases and coverage rate of the test cases) of the program testing abilities of a more collective set of recent LLMs for code (including 4 LLMs whose sizes are around 1B and 7 LLMs that are significantly larger). We have revised the statement in the paper to avoid possible over-claiming and misunderstandings. By additionally assessing the correctness of a test case (i.e., whether a test case can accurately reflect the requirements of the instruction in the prompt), our experiments show whether the generated code correctly meets the specified criteria, and, building on this, we can filter the model-generated code with these test cases to enhance the program/code synthesis performance.
>
> **Q2:** Two datasets were used in the evaluation. The selection of datasets seems to be arbitrary. What are their characteristics? Are they sufficient? Why are they representative? What will happen to programming languages other than Python? The metrics for evaluating the experimental results cannot be taken as granted.
>
> **A:** When selecting the dataset, we opted for evaluation datasets widely utilized in the current mainstream program/code synthesis tasks. Considering that the primary evaluation of code models is conducted on Python language datasets, to save computational resources without sacrificing the representativeness of their evaluation, we also chose the prominent Python evaluation datasets, i.e., HumanEval and MBPP. These two datasets have been extensively used in the evaluation of many code language models [1][2][3]. We are more than glad to incorporate evaluation on more languages and more datasets if suggested specifically, to make the evaluations more comprehensive.
>
> **Q3:** The pass rate indicates the ratio of tests that pass unit testing, but it cannot tell if a test is good or not and often those tests that do not pass are regarded as more important as the aim of testing is to find defects in the code.
>
> **A:** The pass rate of test cases are calculated with the oracle code \textbf{which is regarded as correct implementations of the programming problems (on HumanEval+ and MBPP) with no bugs}. In fact, the oracle code on HumanEval+ has been tested using fuzzing to ensure their correctness, thus make the assumption more realistic [4]. As mentioned by Reviewer EZ26, we demand generated test case to show both high coverage and "true" intended input-output relationships. In our paper, a test case is considered as correct if the oracle code could pass the test case (i.e., the output produced by the oracle given the test input matches the test output), and it is considered as incorrect otherwise. That is, a higher pass rate indicates that the oracle code passes more on the test cases generated by an LLM, which means this LLM excels more on the task of generating correct test cases that reflecting the "true" intended input-output relationship of the programming problem.
>
> **Q4:** Code coverage is definitely more interesting than pass rate. The code coverage in the results look mediocre. Also, the computation of code coverage in Eq. (2) seems to be wrong. Different tests can cover the same branches and they are aggregated with considering the redundancy in Eq. (2). As a result, the final coverage might be over-estimated.
>
> **A:** We evaluate not only the pass rate but also the code coverage of LLM-generated test cases, such that we test not only if the generated test cases represent "true" desired input-output relationship of the described programming problem, but also how much the test cases covers the code for identifying bugs. As for the way of calculating the coverage rate, we would like to make it clearer that we calculate the average per-test-case branch coverage, instead of aggregating the coverage rates of different test cases. That said, there is no over-estimation of the coverage rate.

---

> ### Author Response · Authors · 2023-11-21
>
> **Q5:** I do not understand what the following statement means: “the test cases generated by LLMs can show a descent pass rate, and this pass rate is even higher than the code pass rate on HumanEval+, which holds for both large and small LLMs.” (Page 5)
>
> **A:** This statement means that the correctness of the generated test cases is even higher than the correctness of code implementations for problems on HumanEval+. As explained in our response to Q3, the pass rate (of a test case) indicates the correctness of the test case and shows how well it represents the desired input-output relationships of the programming problem on HumanEval+ and MBPP. Thus, a higher pass rate (than the code pass rate) of the test cases shows that the generated test cases, in general, are more representative of the programming problem than the generated programs/code. This conclusion holds true for both large and small models.
>
> **Q6:** Existing work already applies LLMs for improving code coverage. What’s the advantage of the approach in this paper? Will it result in on par to even higher coverage? If the answer is no, does it mean that testing ability of LLM exercised in this paper is not the state of the art?
>
> **A:** Please see our response to Q1. Thanks.
>
> **Q7:** What are the principles for dataset selection in the paper?
>
> **A:** Please see our response to Q2. Thanks.
>
> **Q8:** How to evaluate the quality of tests generated by LLMs in this paper?
>
> **A:** Please see our responses to Q3 and Q4. Thanks.
>
> [1] CodeGen: An Open Large Language Model for Code with Multi-Turn Program Synthesis
>
> [2] InCoder: A Generative Model for Code Infilling and Synthesis
>
> [3] StarCoder: may the source be with you!
>
> [4] Is your code generated by ChatGPT really correct? rigorous evaluation of large language models for code generation.

---

> ### Author Response · Authors · 2023-11-23
> **Look forward to your reply**
>
> Dear Reviewer PLSn,
>
> We are grateful for your insightful comments and suggestions, and we have made every effort to address your concerns in our responses and the revised paper. As the final deadline for reviewer-author discussions is fast approaching, we would be most appreciative if you could kindly spare some time to provide any additional feedback or comments you may have on our responses and paper. Your expertise and guidance are valuable to us, and we are eager to ensure that our work meets the highest standards possible. Many thanks.
>
> Warmest regards,
> Authors

---

> ### Comment · Reviewer_PLSn · 2023-11-23
>
> Many thanks for the response. I will keep my score. I still believe this work will benefit from the inclusion of techniques that can improve the code generation performance of LLMs, as there is already a lot of research in this area. Additionally, it would be beneficial to introduce a more diverse range of benchmarks that cover various levels of complexity of programs.

---

> ### Author Response · Authors · 2023-11-23
> **Thanks for the further suggestions**
>
> Dear Reviewer PLSn,
>
> We would like to express our utmost gratitude for your feedback.
>
> With regard to your further suggestions on incorporating techniques that could improve the code generation performance and considering alternative datasets, we would be most grateful if you could kindly provide **further clarification or specifications**. To the best of our knowledge, the two datasets we have chosen are in line with a series of recent papers testing LLMs (for code) and are the most appropriate for our study of testing algorithmic coding implementations. Moreover, as previously mentioned, we have opted for a single round of querying LLMs to obtain test cases in this paper, which leaves most techniques which are capable of consistently improving the code generation performance unsuitable. Your suggestions are immensely valuable, but we would like to share our considerations about these choices to avoid possible misunderstandings.
>
> Additionally, we would like to kindly inquire whether our rebuttal and revisions have adequately addressed your **initial questions and comments**. Your satisfaction with our rebuttal and revisions is of utmost importance to us, and we would be more than happy to make any further adjustments if necessary.
>
> Once again, thank you for your time and suggestions, and we hope our responses can address any doubts you may have.
>
> Best regards,
> Authors

---

> > ### Comment · Reviewer_PLSn · 2023-11-23
> >
> > This might be a useful one - "Large Language Models for Software Engineering: Survey and Open Problems" https://arxiv.org/pdf/2310.03533.pdf

---

### Official Review · Reviewer_hLEw · 2023-10-30

**Soundness:** 3 good
**Presentation:** 3 good
**Contribution:** 2 fair
**Rating:** 5
**Confidence:** 4

**Summary:**

The paper presents a thorough experimental evaluation of 11 LLMs for generating tests in 4 different settings:
1. Oracle
2. Self-generated
3. All-generated
4. Placeholder

The paper presents 6 observations from these experiments:
1. Given a task from HumanEval+, the pass rate of generated tests is higher than the pass rate of code.
2. The correctness of tests generated by an LLM is positively correlated with the LLMs ability to generate correct code.
3. Using code generated by the LLM as part of the prompt for test generation leads to better results compared to using only a placeholder.
4. Using Oracle code produces tests of higher quality (better coverage) compared to non-oracle solutions.
5. Using correct code in the prompt produces tests of higher quality (better coverage) compared to using incorrect code.
6. The correctness of the tests generated by an LLM decreases with the order of generation (first is best, last is worst).

Finally, the paper presents a simple technique for improving program synthesis performance by:
1. using self-generation of code instead of placeholder code
2. weight-ranking tests based on their order

Using this technique improves the pass rate of HumanEval+ when using GPT3.5-turbo and CodeT.

**Strengths:**

- Impressive evaluation using 11 LLMs and different testing settings. The benchmarks could be improved, but are on par with what is being used by others in the field.

- A solid set of observations, as outlined in my summary. I would argue that some of these are well-known (eg 1, 3).

**Weaknesses:**

- The experimental evaluation and the observations are nice and make sense to share with the community, but the actual contribution of the paper beyond that is relatively minor.
- Using self-generated code (instead of a placeholder) is not exactly a fair comparison when compared to previous work, as it samples the model 2x for each generation. Using the place-holder code was somewhat intended in the original technique so you only generate the code itself once.

**Questions:**

- How do you "assign different weights to the generated test cases based on their order in each generation result"?

- Coverage rates are surprisingly high for a small number of tests, can you give some examples of program/branches that were hard to cover for all models? How many branches are to be covered for programs in these benchmark sets?

- There is no discussion of the prompts used for each model (outside the mention at the end of 4.1). I expected different prompts (other than the Oracle/SG/AG/PH part) to be part of the evaluation.


- Page 2 "converge rate"
- Fig. 2 Caption: "past rate"
- "SG and WR" -> RW, "SR and/or RW" -> SG

---

> ### Author Response · Authors · 2023-11-21
>
> Thanks for your insightful comments! We have fixed the mentioned typos, and, for other comments, please see our responses below.
>
> **Q1:** Using self-generated code (instead of a placeholder) is not exactly a fair comparison when compared to previous work, as it samples the model 2x for each generation. Using the place-holder code was somewhat intended in the original technique so you only generate the code itself once.
>
> **A:** We would like to explain that, in both the "self-generated" setting and the "placeholder" settings, one needs to query the LLM only ONCE for generating the same number of three test cases. The main difference of these two settings, when generating test cases, lies in the prompts used. In the "placeholder" setting, one uses a placeholder as the code implementation in prompts, while in the self-generated setting, we use the code generated by the LLM itself as the prompt. In the program/code synthesis task, one always has to query a LLM to obtain some code implementations for a programming problem (which means these queries are inevitable), and these code implementations are then used to in the prompts when generating test cases in the "self-generated" setting.
>
> **Q2:** How do you "assign different weights to the generated test cases based on their order in each generation result"?
>
> **A:** Given a set of code implementations and another set of test cases for a programming problem, the consensus algorithm of CodeT groups them into matched sub-sets (i.e., consensus sets) and it picks the code in the largest consensus set (i.e., the consensus set with the most code implementations and test cases) to output. We use the self-generation technique to provide test cases for the algorithm, which means, instead of using test cases obtained in the "placeholder" setting, we opt for test cases that are generated in the "self-generated" setting in order to benefit from their higher quality. As for the rank-based reweighting, we use the rank of each generated test case to reweight its contribution to the consensus set it belongs to. That is, for test cases in a consensus set, instead of directly using their counting numbers, we use the sum of $p^{i-1}$ ($i$ is the order of each test case in the model output and $p=0.8$) when measuring the score. The final score of a consensus set is thus the sum of a) $\sum p^{i-1}$ and b) the number of code implementations in the consensus set, and code implementations in the consensus set with the highest score are considered as the best solutions.
> We have revised our paper to make this part clearer, providing a more comprehensive overview of the problem.
>
> **Q3:** Coverage rates are surprisingly high for a small number of tests, can you give some examples of program/branches that were hard to cover for all models? How many branches are to be covered for programs in these benchmark sets?
>
> **A:** For example, on HumanEval+, the coverage rate of code for the problem #46 is below 50%. The instruction for this problem is ```def fib4(n: int):    \"\"\"The Fib4 number sequence is a sequence similar to the Fibonacci sequence that's defined as follows:\n    fib4(0) -> 0\n    fib4(1) -> 0\n    fib4(2) -> 2\n    fib4(3) -> 0\n    fib4(n) -> fib4(n-1) + fib4(n-2) + fib4(n-3) + fib4(n-4).\n    Please write a function to efficiently compute the n-th element of the fib4 number sequence.  Do not use recursion.   \"\"\"\n```.  And the oracle solution is ```    if n == 0:\n        return 0\n    elif n == 1:\n        return 0\n    elif n == 2:\n        return 2\n    elif n == 3:\n        return 0\n    else:\n        a, b, c, d = 0, 0, 2, 0\n        for i in range(4, n + 1):\n            a, b, c, d = b, c, d, a + b + c + d\n        return d\n\n```.
> The average number of branches in code implementations on HumanEval+ is 5.14.
>
> **Q4:** There is no discussion of the prompts used for each model (outside the mention at the end of 4.1). I expected different prompts (other than the Oracle/SG/AG/PH part) to be part of the evaluation.
>
> **A:** As mentioned in Section 5 in the paper, the prompt for generating test cases are given by concatenating the function definitions and docstrings ("def cycpattern_check(a, b): \n \t """...."), the code implementation ("c=a \n ....") or a placeholder ("pass"), and a comment given to prompt test case generation ("# Check the correctness of this function with three test cases..."). In our early experiments, we found that modifying the final comment given to prompt test case generation only has a relatively small impact on the test case pass rate. We have tried e.g., "# Verify if the function is accurate and generate three test cases..." and "# Generate three test data to verify the correctness of this function..." and only observed less than $\pm$0.50% difference in correctness of the obtained test cases. We have discussed the effect of different prompts in the revised appendix.

---

> ### Author Response · Authors · 2023-11-23
> **Look forward to your reply**
>
> Dear Reviewer hLEw,
>
> We are grateful for your insightful comments and suggestions, and we have made every effort to address your concerns in our responses and the revised paper. As the final deadline for reviewer-author discussions is fast approaching, we would be most appreciative if you could kindly spare some time to provide any additional feedback or comments you may have on our responses and paper. Your expertise and guidance are valuable to us, and we are eager to ensure that our work meets the highest standards possible. Many thanks.
>
> Warmest regards,
> Authors

---

### Official Review · Reviewer_7enN · 2023-10-31

**Soundness:** 3 good
**Presentation:** 3 good
**Contribution:** 3 good
**Rating:** 6
**Confidence:** 5

**Summary:**

This paper studies the effectiveness of code models and LLMs in generating test cases for programs. The quality of the generated test cases is evaluated using the pass rate (percentage of generated tests that are valid, i.e., pass when testing a correct implementation) and coverage rate (percentage of branches in the program that are covered by the test suite). The authors evaluate four different types of prompts for generating tests and consider 11 different code models. In order to motivate the use of LLMs for generating tests, it is argued that additional tests can help improve the performance of LLMs for code synthesis tasks. To support this argument, empirical results for code synthesis tasks are presented when using the LLM-generated test cases. The empirical results are promising.

**Strengths:**

1. The ability of LLMs to generate valid and useful test cases is an interesting question worthy of study.

2. The experimental design is largely sound (see some concerns below in **Questions**) and the experiments are comprehensive (in terms of the number of models and datasets considered).

3. The experiments lead to interesting yet expected conclusions. For instance, larger models and models with better code synthesis capabilities also tend to be better at generating test cases. It also turns out to be helpful to provide a correct implementation of the code to be tested when generating useful test cases.

**Weaknesses:**

1. It is not clear to me why test case generation is restricted to three test cases. In practice, one would want to generate a much larger number of tests and therefore, it is important to evaluate the performance of the models in such settings. The authors already note that the first generated test case often is more likely to be correct than the third test. If the number of generated tests is increased, this problem is likely to become worse and this would suggest that LLMs should NOT be used for generating tests.

2. I am not sure what the takeaway message is from the paper. It seems like if one actually intends to use the test cases as part of a test suite, it is perhaps not a good idea to use LLM-generated tests since they are likely to be incorrect. However, for more *fuzzy* use of tests such as providing guidance during LLM-generated code synthesis, such generated tests could be useful. It would help if the paper includes a discussion along these lines.

**Questions:**

I have a number of questions regarding the experiments. I list them below:
1. (Section 2) I assume that the pass rate is calculated with respect to the oracle program. Is this correct? Is the coverage rate also calculated with respect to the oracle program?

2. (Section 4.1) How is the *all-generated* code set created? I understand each LLM is used to generate 100 code completions but how are these combined? Also, why does all-generated code only include code from the smaller models? Finally, what does the all-generated prompt look like exactly?

3. (Section 4.1) Why is the temperature set 0.2 for all code generation? Why not use the temperature setting that is specified in the papers for each model?

4. (Section 4.3) Are the code comments used in the "Placeholder" setting also provided in the other settings?

5. (Table 6) For calculating the Pass@1 rate, are the synthesized tests also considered?

---

> ### Author Response · Authors · 2023-11-21
>
> Thanks for your insightful comments! Please see our reponses below.
>
> **Q1.** It is not clear to me why test case generation is restricted to three test cases. In practice, one would want to generate a much larger number of tests and therefore, it is important to evaluate the performance of the models in such settings. The authors already note that the first generated test case often is more likely to be correct than the third test. If the number of generated tests is increased, this problem is likely to become worse and this would suggest that LLMs should NOT be used for generating tests.
>
> **A:** The decision to limit the test cases to three was made to strike a balance between practicality and effectiveness of the study. By limiting the model to produce a fix number (e.g., 3) of test samples in the prompt, we aim to assess and compare the testing capabilities of the LLMs in a single run of generation. Increasing the number of generated tests of generating test cases in an unconstrained setting is an interesting point for future research, and we believe that our study can act as a seminal study to provide valuable insights into the potential of LLMs in generating effective initial test cases.
> Also, we would like to stress that the aim of this work is to study program testing ability of LLMs for code (i.e., CodeX, StarCoder, CodeT5+, etc.), instead of advocating using LLMs as generator of test cases for any applications. Although the quality of later test cases decreases with when more test cases are generated, there exist other ways of keep the high quality of test cases when generating more ones (e.g., by querying LLMs multiple times with a moderately large temperature). As mentioned by Reviewer EZ26, we demand generated test case to show both high coverage and "true" intended input-output relationships. Symbolic execution and evolutionary fuzzing excel in achieving the first goal, though they do not guarantee the latter. By contrast, LLMs show the potential to represent "true" intended input-output relationships reasonably, yet they are not explicitly asked to maximize coverage. We feel that it is promising to explore possible combinations of these techniques in the future (e.g., making LLMs as a generator of the fuzzing seeds, etc), and we believe this paper (as a seminal study of the programing testing abilities of these LLMs from both perspectives) will also inspire new research ideas in this direction.
> We have revised the paper accordingly to make these points clearer.
>
> **Q2:** I am not sure what the takeaway message is from the paper. It seems like if one actually intends to use the test cases as part of a test suite, it is perhaps not a good idea to use LLM-generated tests since they are likely to be incorrect. However, for more fuzzy use of tests such as providing guidance during LLM-generated code synthesis, such generated tests could be useful. It would help if the paper includes a discussion along these lines.
>
> **A:** We primarily investigated the ability of current LLMs for code in generating test cases and their related characteristics. Although test cases generated by current LLMs are not perfectly correct, note that these models are still rapidly evolving, and thus the quality of generated test cases will also be continually improved from our perspective. Thus, we believe that the studies of these models in generating test cases is worthwhile and inspiring to the community. Moreover, as mentioned, we have actually demonstrated intriguing observations about LLMs in generating test cases in this paper, which are also utilized to enhance their program/code synthesis performance and improve understanding of these LLMs. We have revised the paper accordingly to make this clearer.
>
> **Q3.** (Section 2) I assume that the pass rate is calculated with respect to the oracle program. Is this correct? Is the coverage rate also calculated with respect to the oracle program?
>
> **A:** Yes, the coverage is also calculated with respect to the oracle program, just to make the coverage rate evaluated in different settings being comparable. We provide results about coverage rates calculated with respect to the tested code (in the prompt) in the appendix.

---

> ### Author Response · Authors · 2023-11-21
>
> **Q4.** (Section 4.1) How is the all-generated code set created? I understand each LLM is used to generate 100 code completions but how are these combined? Also, why does all-generated code only include code from the smaller models? Finally, what does the all-generated prompt look like exactly?
>
> **A:** We used four LLMs whose sizes are around 1B (i.e., InCoder 1.3B, CodeGen2 1B, CodeT5+ 770M, SantaCoder 1.1B) to synthesize the "All-generated" code. For each problem, we let each of the four LLMs generate 100 program completions, using their official prompt for program synthesis/completion, and we randomly sampled 25 synthesized implementation from the 100 generated completions to form the "all-generated" dataset. That is, each LLM provides 25 implementations, and thus we have 25x4=100 code implementations to be tested in the "all-generated" setting.
> When generating code to be tested in the "all-generated" setting, we concatenate the function definitions and docstrings ("def cycpattern_check(a, b): \n \t """...."), the generated code ("c=a \n ...."), and a comment given to prompt test case generation ("# Check the correctness....") to encourage generating test cases. Please see Figure 1 in the paper for an example.
>
> **Q5.** (Section 4.1) Why is the temperature set 0.2 for all code generation? Why not use the temperature setting that is specified in the papers for each model?
>
> **A:** When evaluating the code pass rate (pass@1), we set the inference temperature as 0.2 by following the settings in previous papers [1][2][3][4]. In addition, we keep the same temperature as a way to control variables in experiments.
>
> **Q6.** (Section 4.3) Are the code comments used in the "Placeholder" setting also provided in the other settings?
>
> **A:** Yes, the the docstrings are also given in other settings when generating test cases.
>
> **Q7.** (Table 6) For calculating the Pass@1 rate, are the synthesized tests also considered?
>
> **A:** When evaluating the code pass rate (pass@1), we only consider the program/code synthesis results and the test are not generated yet at this stage.
>
> [1] CodeT5+: Open Code Large Language Models for Code Understanding and Generation
>
> [2] CodeGen: An Open Large Language Model for Code with Multi-Turn Program Synthesis
>
> [3] InCoder: A Generative Model for Code Infilling and Synthesis
>
> [4] StarCoder: may the source be with you!

---

> ### Comment · Reviewer_7enN · 2023-11-21
>
> I thank the authors for their detailed response. Regarding **Q4**, I think Figure 1b can be slightly confusing. Looking at the figure, it seemed to me that the code completions from different LLMs are included simultaneously in the same prompt but the response helped clarify that this is not the case.

---

> > ### Author Response · Authors · 2023-11-21
> > **Thanks for the prompt reply.**
> >
> > Dear Reviewer 7enN,
> >
> > We would like to thank you for the prompt reply and further suggestions. We have revised Figure 1 accordingly to avoid possible confusions and misunderstandings. If you have any further suggestions or concerns, please do not hesitate to let us know. Thanks again!
> >
> > Best regards,
> > Authors

---

### Official Review · Reviewer_EZ26 · 2023-11-02

**Soundness:** 3 good
**Presentation:** 1 poor
**Contribution:** 2 fair
**Rating:** 3
**Confidence:** 5

**Summary:**

Automated test case generation is an important field of research in software engineering. A piece of code designed to solve a certain problem may have some implementation errors. Given such code, we would like to:

* generate inputs that cover most of the paths in the program (high coverage)
* find the “true” outputs for these inputs that test the _intended_ functionality of the code. Note that this is not the same as simply running each input through the program under test. The program may have errors, and the model has to “see through” the implementation to find the intended functionality. This is called the oracle problem.

Both these problems have been extensively studied in the software engineering community. The first problem is typically solved with a combination of symbolic execution and evolutionary fuzzing. The second problem (the oracle problem) is much more challenging, and is perhaps an excellent domain to apply language models. The “intended” functionality of code is hard to infer by traditional analysis, but language models can a) use comments/documentation, and b) use semantic properties of the code itself.

This paper attempts to evaluate language models to solve __both__ of the above problems in a single pass.

The language model is shown a natural language description of a programming task, and potentially buggy code solution to that task. Then the model is asked to generate test cases consisting of _both_ inputs and oracle outputs.

The generated test cases are evaluated in two ways - a) code coverage on the presented buggy code solution, b) pass rate on the ground truth oracle solution.

In this paper, the buggy code to be tested is _also code generated by a language model_. The authors consider multiple settings for this:
* one where the code is **self-generated**,
* one where the code is generated by **other models**,
* one where the code is actually oracle code, and
* one where the code is just a placeholder stub and the model has to use only the problem description.

The authors analyze their results in these settings, and then use their insights to solve a related problem - **code synthesis**. Recent work (CodeT) has shown that generating tests along with code and using these tests to filter the generated code can improve generation accuracy. The authors use two insights to improve the performance of CodeT:
* _Large models_ generate better tests (as measured by pass rate on the oracle) when shown their own possibly buggy generated code (self-generation setting), versus when not shown any code at all (placeholder setting).
* If a model generates multiple tests in sequence, then earlier tests in the sequence are more likely to be correct than those that come later.

Finally, the authors use these two insights to improve CodeT, and improve its code generation accuracy by up to 4.2 percentage points.

In summary, the paper has two main contributions - a) an analysis of how well language models can generate tests, and b) using insights from this analysis to improve the program generation accuracy of CodeT.

**Strengths:**

1. First analysis of the test case generation ability of large language models for competitive coding (algorithmic coding) problems.

1. I like the takeaway that a model generates better test cases when you first ask it to generate code to solve the problem (analogous to Chain of Thought). This is a good empirical result backed by intuition.

**Weaknesses:**

I like the general premise of this work. However, it it my opinion that it is not ready for publication in its current form due to several drawbacks:

1. There are two distinct problems that this paper is trying to solve. The first is test generation, and the second is test-guided program generation (analogous to CodeT). However in attempting to do both of these, the paper ends up not doing justice to either one.

1. If the aim is to study test generation with language models, then:
    - The paper needs to include, up-front, a discussion of code coverage, the oracle problem, other traditional software engineering approaches to solve this problem (test generation has seen a long line of work), and even other transformer-based approaches like Dinella et. al (2022) [1]. I understand that not all of these are relevant to competitive/algorithmic coding tasks, but there should be a thorough discussion and setting up of the problem to motivate your approach.
    - Buggy code generated by language models can’t be the _only_ setting in which test generation is evaluated. There are other examples of buggy code, for example, CodeNet includes several incorrect human solutions for each problem. It would be interesting to see how test generation works for these.
    - The language model is never explicitly asked to or instructed to maximize coverage while generating tests. In fact, good coverage is almost an accidental by-product of the process. I could think of multiple ways to prompt the model to achieve better coverage, possibly with iterative refinement. And then there’s the question of whether a language model is actually the best choice to maximize coverage at all, or whether a fuzzer could do better.

1. If the aim is to improve test-guided program generation by generating better tests, then:
    - The description of the methodology is so brief that it’s difficult to fully understand how the authors build on CodeT. For example - how/where is the re-weighted score used? From my understanding, CodeT forms symmetry groups based on which solutions pass the _same sets_ of test cases (like a consensus). I don’t understand how one would use re-weighting here. Similarly, when it comes to tests using self-generated code, how exactly do you incorporate this in the consensus algorithm of CodeT?
    - The empirical gains aren’t particularly striking unfortunately - the gain is limited to 4 percentage points for the largest model, and is minimal for smaller models. I wonder if some of your other insights could be used to further improve this performance (like you mentioned as a future direction).

1. The writing and organization of this paper leave a lot to be desired. Here is a short list:
    - The introduction is extremely brief and doesn’t motivate the problem well. The part about “4 test case generation settings” comes out of nowhere, and terms like “self-generated”, etc. don’t make sense until you mention that you’re testing code that is produced by language models.
    - In the Evaluation section, you didn’t mention whether Coverage rate is evaluated on the code that’s being tested (self-generated, all-generated, etc), or on the oracle code. I assume it’s the former, but this should be specified.
    - In the “All-generated” setting, each problem is associated with multiple solutions. Do you give all of these simultaneously to the model and ask it to generate tests? Or one at a time, with one set of tests per solution?
    - Typos and notation: a) What is the final subscript $k$ in $Q_{ij} = \{(x_{ijk}, y_{ijk})\}_{\textbf{k}}$? b) Top of page 4 - “HuamnEval”, c) page 5 and page 9 - “descent pass rate”, “achieving descent performance”, d) page 1 - “quality of program synthesise”.

[1] Dinella, Elizabeth, et al. "Toga: A neural method for test oracle generation." Proceedings of the 44th International Conference on Software Engineering. 2022.

**Questions:**

1. Can you provide some more details of how exactly you incorporate self-generation and re-weighting in the consensus algorithm of CodeT?

1. Can you clarify the “All-generated” setting - how exactly do you provide multiple solutions to the language model while generating tests?

---

> ### Author Response · Authors · 2023-11-21
>
> Thank you for your feedback and helpful suggestions for our work. Our answers to your questions and response to the comments are given in details as below.
>
> **Q1:** There are two distinct problems that this paper is trying to solve. The first is test generation, and the second is test-guided program generation (analogous to CodeT). However in attempting to do both of these, the paper ends up not doing justice to either one.
>
> **A:** We appreciate the comment. The aim of this work is to study program testing abilities of LLMs for code (i.e., CodeX, StarCoder, CodeT5+, etc.), considering that 1) program testing is of great interest in software engineering and software security and 2) test cases can further be adopted to improve the program/code synthesis performance, as the tasks of program/code synthesis and program testing are closely correlated. Therefore, though our main focus is to study the program testing abilities of these LLMs, we also discuss 1) how enhanced program testing abilities can be easily obtained and 2) how to appropriately utilize the enhanced testing abilities to further improve the program/code synthesis performance of these LLMs.
> We have revised our paper accordingly to make the aim of this paper clearer.
>
> **Q2:** The paper needs to include, up-front, a discussion of code coverage, the oracle problem, other traditional software engineering approaches to solve this problem (test generation has seen a long line of work), and even other transformer-based approaches like Dinella et. al (2022) [1]. I understand that not all of these are relevant to competitive/algorithmic coding tasks, but there should be a thorough discussion and setting up of the problem to motivate your approach.
>
> **A:** We recognize the necessity of providing a clearer delineation of these objectives and their respective contributions to avoid any confusions and misunderstandings. Thus, following your suggestion, we have revised the introduction section and incorporated discussions on code coverage and traditional software engineering approaches to better articulate the motivation behind our paper.
>
> **Q3:** Buggy code generated by language models can’t be the only setting in which test generation is evaluated. There are other examples of buggy code, for example, CodeNet includes several incorrect human solutions for each problem. It would be interesting to see how test generation works for these.
>
> **A:** We appreciate the suggestion. However, after carefully examining the CodeNet dataset, we found that CodeNet unfortunately lacks oracle code to aid in the analysis of testing the correctness of synthesized code. Additionally, the problem instructions in CodeNet are challenging to be transformed into a code completion format, making it difficult to adapt to our framework for testing competitive/algorithmic coding implementations. For instance, it often only provides a script without any main function, which makes the evaluation of generated tests nontrivial in our framework where the testing code can be obtained by simply completing an assert statement using an LLM. We can focus on coverage rates on this dataset, but this largely limits the other aim of exploring correctness of test cases (which represent "true" intended input-output relationships of a function/program) in this paper.

---

> > ### Author Response · Authors · 2023-11-21
> >
> > **Q4:** The language model is never explicitly asked to or instructed to maximize coverage while generating tests. In fact, good coverage is almost an accidental by-product of the process. I could think of multiple ways to prompt the model to achieve better coverage, possibly with iterative refinement. And then there’s the question of whether a language model is actually the best choice to maximize coverage at all, or whether a fuzzer could do better.
> >
> > **A:** We would like to explain that it is NOT claimed that LLMs are the best options to maximize code coverage. We investigate the ability of LLMs in the task of generating test cases, by examining not only the correctness of test cases generated by LLMs (i.e., the matching degree between the input and output of the test cases) but also the coverage of the test. In experiments, we aim to compare the program testing abilities of different LLMs in different settings, without modifying the architecture of these LLMs, or re-training these LLMs, or significantly increasing the computational burden (which would limit possible usage of many practical application scenarios including program/code synthesis). Thus, although the code coverage can indeed be increased by querying the LLMs multiple times as mentioned, we did not try in this work.
> > As mentioned by the reviewer, we demand generated test case to show both high coverage and "true" intended input-output relationships. Symbolic execution and evolutionary fuzzing excel in achieving the first goal, though they do not guarantee the latter. By contrast, LLMs show the potential to represent "true" intended input-output relationships reasonably, yet they are not explicitly asked to maximize coverage. We feel that it is promising to explore possible combinations of these techniques in the future (e.g., making LLMs as a generator of the fuzzing seeds, etc), and we believe this paper (as a seminal study of the programing testing abilities of these LLMs from both perspectives) will also inspire new research ideas in this direction.
> >
> > **Q5:** The description of the methodology is so brief that it’s difficult to fully understand how the authors build on CodeT. For example - how/where is the re-weighted score used? From my understanding, CodeT forms symmetry groups based on which solutions pass the same sets of test cases (like a consensus). I don’t understand how one would use re-weighting here. Similarly, when it comes to tests using self-generated code, how exactly do you incorporate this in the consensus algorithm of CodeT?
> >
> > **A:** Given a set of code implementations and another set of test cases for a programming problem, the consensus algorithm of CodeT groups them into matched sub-sets (i.e., consensus sets) and it picks the code in the largest consensus set (i.e., the consensus set with the most code implementations and test cases) to output. We use the self-generation technique to provide test cases for the algorithm, which means, instead of using test cases obtained in the "placeholder" setting, we opt for test cases that are generated in the "self-generated" setting in order to benefit from their higher quality. As for the rank-based reweighting, we use the rank of each generated test case to reweight its contribution to the consensus set it belongs to. That is, for test cases in a consensus set, instead of directly using their counting numbers, we use the sum of $p^{i-1}$ ($i$ is the order of each test case in the model output and $p=0.8$) when measuring the score. The final score of a consensus set is thus the sum of a) $\sum p^{i-1}$ and b) the number of code implementations in the consensus set, and code implementations in the consensus set with the highest score are considered as the best solutions.
> > We have revised our paper to make this part clearer, providing a more comprehensive overview of the problem.
> >
> > **Q6:** The empirical gains aren’t particularly striking unfortunately - the gain is limited to 4 percentage points for the largest model, and is minimal for smaller models. I wonder if some of your other insights could be used to further improve this performance (like you mentioned as a future direction).
> >
> > **A:** Thanks for the comment. We would like to argue that, improving the quality of program/code synthesis is very challenging, especially without modifying/re-training the LLM. Our paper mainly studies the program testing ability of LLMs for code and we utilize some intriguing findings to improve the program/code synthesis performance as a byproduct. The benefit of our improvements in the program/code synthesis performance does not come from modifying the structure of a LLM, re-training a LLM, or significantly increasing the number of code samples. Instead, we only leverage the improved quality of test cases. Considering this, we believe that achieving a significant +4% increase in the code pass rate is actually noteworthy.

---

> > > ### Author Response · Authors · 2023-11-21
> > >
> > > **Q7:** The introduction is extremely brief and doesn’t motivate the problem well. The part about “4 test case generation settings” comes out of nowhere, and terms like “self-generated”, etc. don’t make sense until you mention that you’re testing code that is produced by language models.
> > >
> > > **A:** We appreciate the comment and have revised the introduction section accordingly.
> > >
> > > **Q8:** In the Evaluation section, you didn’t mention whether Coverage rate is evaluated on the code that’s being tested (self-generated, all-generated, etc), or on the oracle code. I assume it’s the former, but this should be specified.
> > >
> > > **A:** Yes, the coverage rate is evaluated on the code that's being tested. We have revised the paper accordingly to make it clearer.
> > >
> > > **Q9:** Can you provide some more details of how exactly you incorporate self-generation and re-weighting in the consensus algorithm of CodeT?
> > >
> > > **A:** Thanks for the question. Please see our response to Q5 for all the details.
> > >
> > > **Q10:** Can you clarify the “All-generated” setting - how exactly do you provide multiple solutions to the language model while generating tests?
> > >
> > > **A:** We used four LLMs whose sizes are around 1B (i.e., InCoder 1.3B, CodeGen2 1B, CodeT5+ 770M, SantaCoder 1.1B) to synthesize the "All-generated" code. For each problem, we let each of the four LLMs generate 100 program completions, using their official prompt for program synthesis/completion, and we randomly sampled 25 synthesized implementation from the 100 generated completions to form the "all-generated" dataset. That is, each LLM provides 25 implementations, and thus we have 25x4=100 code implementations to be tested in the "all-generated" setting.

---

> ### Author Response · Authors · 2023-11-23
> **Look forward to your reply**
>
> Dear Reviewer EZ26,
>
> We are grateful for your insightful comments and suggestions, and we have made every effort to address your concerns in our responses and the revised paper. As the final deadline for reviewer-author discussions is fast approaching, we would be most appreciative if you could kindly spare some time to provide any additional feedback or comments you may have on our responses and paper. Your expertise and guidance are valuable to us, and we are eager to ensure that our work meets the highest standards possible. Many thanks.
>
> Warmest regards,
> Authors